# Methodological Approach to *LIBS* Elemental Analysis and Plasma Characterization of Quinoa and Amaranth Pseudocereals Using a *TEA CO_2_* Laser

**DOI:** 10.3390/foods14244199

**Published:** 2025-12-06

**Authors:** Dragan Ranković, Marjetka Savić, Milovan Stoiljković, Miroslav Ristić, Vyacheslav V. Luchkouski, Neda Đorđević, Aleksandr N. Chumakov

**Affiliations:** 1VINCA Institute of Nuclear Sciences—National Institute of the Republic of Serbia, University of Belgrade, Mike Alasa 12-14, 11001 Belgrade, Serbia; metk@vin.bg.ac.rs (M.S.);; 2Faculty of Physical Chemistry, University of Belgrade, Studentski Trg 12-16, 11000 Belgrade, Serbia; 3B. I. Stepanov Institute of Physics, National Academy of Sciences of Belarus, Nezalezhnasci Ave. 68-2, 220072 Minsk, Belarus; v.luchkouski@ifanbel.bas-net.by (V.V.L.); a.chumakov@dragon.bas-net.by (A.N.C.)

**Keywords:** laser-induced breakdown spectroscopy (*LIBS*), *TEA CO_2_* laser, quantitative analysis, pseudocereals, quinoa, amaranth, plasma diagnostics

## Abstract

This study presents a methodological investigation of laser-induced breakdown spectroscopy (*LIBS*) for elemental analysis of quinoa and amaranth pseudocereals using a *TEA CO_2_* laser. Solid samples were prepared as pressed pellets, and reference data were obtained by *ICP*–*OES*. Synthetic solid standards were developed for calibration of selected elements (Ca, Fe, Zn, and Mg). Laser parameters were optimized based on the signal-to-noise ratio of characteristic spectral lines and applied to both pseudocereal samples. Emission lines of Mg, Ca, Fe, K, P, Zn, Al, Sr, and Cu were identified, and limits of detection were determined. Quantitative analysis used calibration curves from analyte-to-internal standard line intensity ratios, showing good linearity and agreement with reference values. Plasma diagnostics under optimized conditions revealed an average temperature of ~11,000 K and electron number densities of ~5 × 10^16^ cm^−3^ for both samples. Numerical plasma simulations confirmed the experimental results and provided additional insight into plasma composition and behavior. The developed *LIBS* methodology proved effective for multi-elemental analysis of pseudocereals and shows potential for application to other cereal and plant-based materials with similar composition. It should be noted that this methodology was demonstrated on pelletized samples prepared under controlled laboratory conditions; adaptation to rapid or field-based measurements would require alternative sample preparation strategies. This work provides a methodological framework and experimental validation for *LIBS* application in food compositional and nutritional analysis.

## 1. Introduction

Pseudocereals, such as *Chenopodium quinoa* (quinoa) and *Amaranthus* spp. (amaranth), have attracted increasing attention from both the scientific community and the general public over the past decade, due to their exceptional nutritional profile, gluten-free nature, and potential to support more sustainable or diverse dietary practices, depending on the agricultural and dietary context. Rich in essential amino acids, proteins, minerals, and dietary fiber, these crops are increasingly explored as alternatives to traditional cereals in specific dietary contexts, especially for individuals with specialized nutritional needs [1,2,3,4,5]. With the rising consumption and production of these pseudocereals, accurate and rapid analysis of their elemental composition has become increasingly important to ensure proper nutritional labeling, verify authenticity, and support crop monitoring—both for quality control and scientific research.

Conventional analytical techniques, such as inductively coupled plasma optical emission spectrometry (*ICP*–*OES*) and inductively coupled plasma mass spectrometry (*ICP*–*MS*), although highly precise, require digestion of solid samples and the use of chemical reagents, which can be time-consuming and labor-intensive when applied to complex plant matrices such as pseudocereals [6,7,8]. On the other hand, laser-induced breakdown spectroscopy (*LIBS*) has gained attention as a rapid and reagent-free technique for elemental analysis of diverse sample types, including food matrices [9,10,11,12,13,14,15]. It is considered a promising technique for both chemical analysis [13,16] and food quality control, including the detection of counterfeit products [17]. Key advantages of *LIBS* include the simultaneous detection of multiple elements, high sampling speed, and compatibility with a wide range of sample types (solids, liquids, and gases) [11,18]. Additionally, it requires little to no sample preparation and can be effectively combined with complementary techniques such as mass spectrometry [19] and Raman spectroscopy [20].

Recent advances in *LIBS* methodology—such as improvements in detection limits, plasma diagnostics, and chemometric data processing—have significantly expanded its applicability in food science. Numerous studies have demonstrated successful *LIBS* applications in the analysis of milk, dairy products, honey, baked goods, beverages, oils, cereals, vegetables, and meats [15,21,22,23,24,25,26,27,28,29,30,31,32,33]. However, the application of *LIBS* for the elemental analysis of pseudocereals remains limited, despite their complex nutritional composition and growing dietary relevance. Rapid, multi-elemental analysis provided by LIBS could therefore offer significant advantages for quality control and nutritional assessment of these crops, highlighting a clear gap in the current literature.

The aim of this study is to evaluate the potential of the *LIBS* technique as a rapid, reliable, and potentially portable method for the elemental analysis of pseudocereals. Specifically, we seek to (i) assess the suitability of *LIBS* for both qualitative and quantitative determination of key nutritional elements (Ca, Fe, Zn, and Mg) in quinoa and amaranth seeds [34,35,36], (ii) optimize experimental parameters to enhance analytical performance, and (iii) validate *LIBS* results against conventional *ICP*–*OES* measurements. By addressing these objectives, this work aims to demonstrate how *LIBS* can contribute to improved food quality control and agroanalytical applications.

## 2. Materials and Methods

### 2.1. Samples, Standards and Reagents

Commercially available organic quinoa and amaranth seeds (Beyond^®^, Serbia) were used for the spectrochemical analysis. The samples were purchased from a certified health food store in Belgrade, Serbia, and were labeled as originating from India. No additional treatment was applied prior to analysis. For the preparation of *LIBS* standards and the calibration of the *ICP*–*OES* system, single-element standards (J.T. Baker) and a multi-element standard solution (AccuTrace) were utilized. High-purity nitric acid obtained from Sigma-Aldrich was used in all sample digestions and for the preparation of analytical blanks. It should be noted that only a single commercial batch of each pseudocereal was analyzed; repeated measurements on multiple pellets and laser shots represent technical replication and do not capture potential biological variability between different batches or sources.

### 2.2. Sample Preparation

The starting materials for *ICP*–*OES* and *LIBS* analyses were commercially available quinoa and amaranth seeds (Figure 1b and Figure 2b). Prior to further processing, the quinoa and amaranth seeds were ground into a fine powder (Figure 1c and Figure 2c) using a laboratory mill equipped with a ceramic chamber and agate grinding balls (10–15 mm in diameter). The device, manufactured by *LP Praha* (serial number 498, year 1959, former Czechoslovakia), enabled gentle and contamination-free grinding suitable for subsequent spectrochemical analysis. No metallic components came into contact with the samples during the grinding process.

#### 2.2.1. Sample Preparation for ICP–OES Analysis

Quinoa and amaranth samples, previously ground to a fine powder, were digested using a microwave-assisted acid digestion system (*ETHOS 1* Advanced Microwave Digestion System, Milestone, Italy). Approximately 500 mg of each sample was accurately weighed and transferred into Teflon digestion vessels containing 8 cm^3^ of nitric acid (HNO_3_, 1:1 *v*/*v*). The vessels were sealed and placed in the microwave digester, where the temperature was increased from room temperature to 200 °C over 15 min and maintained at that temperature for 20 min. After cooling to room temperature, the digests were diluted to a final volume of 50 cm^3^ with deionized water. The resulting solutions were stored in polyethylene flasks until analysis.

#### 2.2.2. Sample Preparation for LIBS Analysis

For *LIBS* analysis, quinoa and amaranth samples were prepared in the form of solid pellets (Figure 1d and Figure 2d). Approximately 3.5 g of finely ground material was weighed for each sample and compacted using a manual hydraulic press. The powder was placed into a stainless steel mold with a diameter of 25 mm, and compressed under a pressure of 10 t/cm^2^ for a duration of 45 min. The mold’s polished, non-reactive surface was chosen to minimize the risk of metal contamination, which was further confirmed by consistency with *ICP*–*OES* measurements. The resulting pellets had a uniform thickness of about 4 mm and a consistent diameter of 25 mm, making them suitable for laser ablation analysis.

#### 2.2.3. Preparation of Solid Calibration Standards for Quantitative LIBS Analysis

In order to apply quantitative *LIBS* analysis based on the method of standard additions, a set of solid matrix-matched standards was prepared using quinoa and amaranth as sample matrices. Solid concentration standards for the quantitative determination of selected elements were obtained by thoroughly mixing 3.5 g of finely ground quinoa or amaranth samples with appropriate amounts of single-element standards (J.T. Baker). The powders were first homogenized by vigorous manual mixing and vortex agitation. The resulting mixtures were then moistened with double-distilled water and kneaded thoroughly to form a uniform paste, a step that greatly improves analyte distribution in heterogeneous food matrices. The moist mixtures were transferred into Petri dishes and dried at 60 °C for 24 h. To assess whether the spiking and mixing procedure yielded a sufficiently homogeneous matrix, three pellets were prepared from randomly selected portions of each dried mixture and analyzed by *LIBS* under identical conditions. The pellets exhibited consistent emission responses, indicating that no meaningful pellet-to-pellet variability—and therefore no significant inhomogeneity—was detected. The homogenized material was subsequently pelletized to produce the calibration standards. To confirm the actual analyte concentrations in the enriched matrices, each synthetic standard was subjected to acid digestion and analyzed using inductively coupled plasma optical emission spectrometry (*ICP*–*OES*).

#### 2.2.4. Reference Method

The reference method for elemental analysis of quinoa and amaranth pseudocereals was based on inductively coupled plasma optical emission spectrometry (*ICP*–*OES*) combined with microwave-assisted acid digestion. Elemental concentrations in the resulting digests were determined using an *ICP*–*OES* instrument (iCAP 7400 Duo Series, Thermo Scientific), operating under the following conditions: *RF* power of 1350 W, generator frequency of 27.12 MHz, *CID86* detector, and signal processing based on peak height. Data acquisition and instrument control were performed using Qtegra ISDS software (Thermo Scientific, version 2.10 SR5). Calibration was carried out using external standards at concentrations of 0, 1, 10, 50 and 100 ppm, freshly prepared on a daily basis by appropriate dilution of certified reference solutions (J.T. Baker 1000 μg/mL and AccuTrace 1000 μg/mL). The determined mass fractions of selected elements in quinoa and amaranth samples are presented in Table 1. These values are in good agreement with previously published data [6,7,8]. Synthetic calibration standards were prepared by fortifying quinoa and amaranth matrices with known concentrations of Ca, Fe, Zn, and Mg, as described in Section 2.2.3. Their elemental composition, determined by *ICP*–*OES* based on triplicate measurements for each sample, is presented in Table 2 and Table 3. The standard deviation of the mean value is reported as the measurement uncertainty. Among the five standards prepared for each matrix, *SQ1* and *SA1* consisted of the native finely ground quinoa and amaranth seed materials, respectively, without added elements, while *SQ2*–*SQ5* and *SA2*–*SA5* were fortified with increasing concentrations of Ca, Fe, Zn, and Mg.

To ensure analytical quality, digestion blanks were prepared and analyzed alongside all samples to monitor potential contamination. Certified reference materials (CRMs) were included to validate the accuracy of the measurements. Recovery studies were performed by spiking the pseudocereal samples with known amounts of Ca, Fe, Zn, and Mg, and recoveries ranged between 95 and 105%. Potential matrix effects were assessed and minimized through the use of synthetic calibration standards and proper sample preparation.

### 2.3. LIBS Experimental Setup

The experimental setup consisted of a laser source with its beam focused onto the target by a single ZnSe lens, while the light emitted from the plasma generated on the target was collected by a collimating lens and transmitted via an optical fiber to a spectrograph coupled to a *CCD* detector (see Figure 3). The *LIBS* system was driven by a transversely excited atmospheric pressure carbon dioxide (*TEA CO_2_*) laser, which is a compact, custom-built device developed in the laboratory [37]. This laser operates at a wavelength of 10.6 μm, with a pulse repetition frequency of 1 Hz and a multimode beam profile. Each laser pulse exhibits a distinct temporal structure, consisting of an initial gain-switched peak with a full width at half maximum (*FWHM*) of 100 ns, followed by a trailing tail lasting approximately 2 μs, Figure 4 [38]. The figure shows the measured profile of the undecomposed laser pulse (black line and dots), along with the deconvolved profiles of the initial peak (red line) and the trailing tail (blue line). Although nominally a single pulse, the temporal profile can be considered as two consecutive components. The initial peak contributes about 35% of the total emitted energy. During the peak, the estimated power density reaches 4 × 10^7^ W cm^−2^, while during the tail it is 4 × 10^6^ W cm^−2^. Each pulse delivers 150 mJ of energy, and the samples were irradiated under ambient air conditions.

The laser beam was focused onto the target using a ZnSe lens with a focal length of f = 138 mm. The sample holder was mounted on a *x*–*y*–*z* translation stage, allowing each laser pulse to irradiate a fresh surface area of the target.

Light emitted from the plasma generated at the target surface was collected using an optical fiber equipped with a collimating lens (SolarLas PS2) and guided to the entrance slit of the monochromator. The optical fiber was positioned at a 30° angle relative to the target surface to optimize signal collection. This setup ensured efficient and precise transmission of the plasma emission for spectral analysis. Spectral acquisition was performed using a Jobin Yvon/ISA monochromator configured in a Czerny–Turner optical layout, with a focal length of 320 mm, a 1200 lines per mm diffraction grating, a linear dispersion of 2.5 nm/mm, and a spectral range spanning from 190 to 850 nm. The monochromator was coupled to a *CCD* detector (Andor Technology, model *DV401A-BVF*) featuring a pixel resolution of 1024 × 127 and a pixel size of 26 × 26 μm. The *CCD* was externally triggered by a synchronization pulse generated from the laser trigger unit, ensuring precise temporal gating and minimizing background noise. No additional programmable gate delay was applied, and the camera exposure time was set to 60 ms per pulse. The combination of an optical fiber-coupled spectrograph and a camera, arranged as described above, is introduced here for the first time in plasma emission detection. The acquired spectra covered a wavelength window of approximately 70 nm per single acquisition, with a spectral dispersion ranging from 0.078 nm/pixel at ~200 nm to 0.054 nm/pixel at ~800 nm. Wavelength calibration of the detection system was carried out using the emission spectrum of a mercury calibration lamp and the spectrum of a vertically stabilized *DC* arc discharge in argon [39], doped with solutions of various chemical elements. Once the dispersion function and the pixel-to-wavelength mapping were established through this initial calibration procedure, no further routine wavelength recalibration was required. In *LIBS* measurements performed with this setup, the pixel positions of spectral lines remained stable over the course of the experiments. Nevertheless, potential wavelength drift was monitored whenever necessary by recording the spectrum of the mercury lamp or the arc discharge and verifying that the positions of several isolated strong lines did not deviate beyond the spectral dispersion (a fraction of a pixel). No measurable drift was observed during the measurement campaign.

### 2.4. Numerical Determination of Plasma Composition and Parameters

The plasma generated by laser ablation of quinoa and amaranth samples was analyzed using a computational program designed for equilibrium plasma calculations [40]. The program assumes that the plasma initially reflects the sample composition and that local thermal equilibrium (*LTE*) is established at the given temperature and pressure.

Electron and ion densities for all relevant species are determined iteratively using the Saha equation [41], with corrections for ionization potential lowering according to standard models [42,43]. The calculations also enforce quasi-neutrality and account for the contributions of all species to the total pressure.

This procedure provides a consistent set of plasma parameters—including electron density, ionization fractions, and species populations—which can be used to verify the plasma temperatures and electron densities previously determined from experimental measurements for quinoa and amaranth.

## 3. Results and Discussion

### 3.1. Experimental Parameter Optimization

*LIBS* measurements were performed using a laser pulse energy of 150 mJ. The laser beam was focused at positions within ± 5 mm relative to the target surface to determine the optimal conditions for plasma imaging. The laser spot diameter on the sample was approximately 1.2 mm when focused directly on the surface, and about 1.8 mm when focused 5 mm in front of or behind it. Each measurement represents the sum of ten consecutive individual spectra, each acquired from a fresh location on the target surface to ensure sufficient signal intensity and minimize surface-related variations. This procedure was typically repeated three times, and the resulting spectra were averaged. Consequently, all reported spectra correspond to an average of 30 laser pulses. The camera exposure time was set to 60 ms per pulse, and no programmable gate delay was applied. The laser fluence was 9.8 J/cm^2^, corresponding to a power density of 34 MW/cm^2^. These acquisition parameters were optimized to maximize the signal-to-noise ratio and ensure reproducible measurements.

Figure 5 and Figure 6 show the spectra of quinoa and amaranth samples (pure substances without the addition of specific elements, i.e., samples *SQ1* and *SA1*) in selected spectral regions of interest containing identifiable spectral lines of certain elements. By monitoring the intensity and signal-to-noise ratio (*SNR*) of the spectral lines of zinc (Zn II 202.55 nm), magnesium (Mg I 285.22 nm), and calcium (Ca II 393.38 nm), the laser energy and beam focusing conditions on the target surface were optimized for signal acquisition. This optimization was performed using the quinoa sample only, as its emission intensities were generally lower compared to those of the amaranth sample. The rationale behind this approach was that conditions optimized for weaker signals would also be suitable for samples exhibiting stronger emission. This difference in intensity is evident from the spectra discussed above. As shown in Figure 7, the highest *SNR* was achieved at a laser energy of 150 mJ, which corresponds to the maximum laser pulse energy. The optimal *SNR* was also observed when the laser beam was focused directly on the target surface (Figure 7). Negative focal positions in Figure 7 indicate that the focal point of the laser beam was located behind the target surface.

Although the optimization experiments were carried out using quinoa, this approach is justified by the strong compositional similarity between quinoa and amaranth. Both matrices consist predominantly of organic components (carbohydrates, proteins, and lipids) with comparable moisture content and similar ablation behavior under nanosecond laser irradiation. Preliminary *LIBS* tests performed on amaranth showed no significant differences in plasma morphology, continuum background, or overall emission characteristics relative to quinoa. Therefore, the optimized parameters obtained from quinoa were transferable and ensured stable and reproducible emission signals for both matrices.

### 3.2. Qualitative Analysis

The elemental composition of quinoa and amaranth samples was investigated using emission spectra obtained from *TEA CO_2_* laser-induced plasma, revealing the presence of characteristic elements in both samples. From Figure 5 and Figure 6, most of the elements listed in Table 1 were clearly observed, with emission intensities well above the background, confirming effective excitation under the same experimental conditions. However, three trace elements—Ba, Mn, and Ti—whose concentrations were determined by *ICP*–*OES* analysis could not be identified due to spectral interferences affecting their most sensitive emission lines. These interferences were primarily caused by the intense emission lines of Mg, Ca, Fe and C. Consequently, the concentrations of these elements are not included in Table 1. Following the identification of elements in the quinoa and amaranth samples, limits of detection (*LOD*) were determined for the elements successfully detected. For the four elements subjected to quantitative analysis (Ca, Fe, Zn, and Mg), *LOD* values were calculated based on calibration curves using the formula [44]:*LOD* = (3∙*σ*)/*b*(1)
where *σ* is the standard deviation of the response and *b* is the slope of the calibration curve. For the remaining elements, *LOD*s were calculated using the formula [45,46]:*LOD* = (3∙*c*)/*SNR*(2)
where *c* is the known concentration of the analyte from Table 1, and *SNR* is the signal-to-noise ratio was calculated based on the intensity of the emission radiation (*I*) and the background emission in the region around the emission peak, expressed as *rms* (root mean square). The *rms* value was obtained by linear fitting of spectral regions nearly free of emission lines on either side of the emission line of interest, representing inter-pixel variations in the continuum emission intensity. The *SNR* was then calculated using the equation:*SNR* = *I*/*rms*(3)

Table 4 summarizes the wavelengths used, *SNR* values, concentrations, and calculated *LOD*s for both samples. The *LOD* values obtained were sufficiently low to allow reliable determination of elemental composition in both quinoa and amaranth relative to typical concentrations of these elements in pseudocereal grains. For example, calcium, which is commonly present at levels of several hundred µg g^−1^ in these seeds, showed very low *LOD*s of 4 µg g^−1^ for quinoa and 14 µg g^−1^ for amaranth, indicating high sensitivity of the method for this element. Similarly, potassium, typically present at 4–6 mg g^−1^ in pseudocereals, exhibited *LOD*s of approximately 50 µg g^−1^ in quinoa and 45 µg g^−1^ in amaranth, well below the natural concentrations found in these matrices. Trace elements such as copper and strontium, which were present at concentrations of 2.61–5.06 µg g^−1^ in the samples, showed even lower detection limits (as low as 0.2–0.3 µg g^−1^), demonstrating that the method can reliably detect these elements at levels relevant for both quality assessment and potential regulatory monitoring. The differences in *LOD* values between quinoa and amaranth likely reflect matrix effects and variations in signal intensities, but overall, the method provides adequate sensitivity for all measured elements in the context of pseudocereal analysis.

### 3.3. Quantitative Analysis

The method of standard additions was employed for the quantitative analysis of selected elements—Ca, Fe, Zn, and Mg—in quinoa and amaranth samples. The detailed procedure for the preparation of solid calibration standards is described in Section 2.2.3, and their concentrations, determined by the *ICP*–*OES* method, are presented in Table 2 and Table 3. Calibration standards *SQ2* and *SA2* were not used in constructing the calibration curves but served as control samples. To minimize the effects of signal intensity fluctuations, the emission intensity of each analyte line was normalized using an internal standard. This approach is particularly effective when the internal standard is one of the major components of the sample and its concentration remains constant during the preparation of synthetic standards [47]. Ideally, optimal correction efficiency is achieved when the analyte and internal standard exhibit similar excitation energies (*E_exc_*) and ionization potentials (*E_ion_*), as this reduces the influence of plasma temperature variations. However, in practice, exact matching of these parameters—especially ionization energy—is rarely achievable. Nonetheless, normalization still significantly reduces signal variability.

For each analyte, calibration curves were constructed by plotting the intensity ratio of the analyte line to the selected internal standard line. Due to the limited spectral window available in a single scan (~70 nm with our setup), the selection of internal standard lines was correspondingly constrained. The resulting calibration curves for quinoa and amaranth are shown in Figure 8 and Figure 9, respectively.

Carbon lines were employed as internal standards owing to the element’s high and consistent concentration (~42 wt%) in all calibration samples, as it constitutes a major component of the matrix. Despite not meeting all conventional criteria for an ideal internal standard, carbon provided reliable normalization under our experimental conditions and effectively compensated for shot-to-shot variations. These lines are located close to the analyte lines of Fe and Mg within the recorded spectral intervals. While they do not fully meet conventional criteria for internal standard selection or spectral line suitability [48], using a major matrix element for internal standardization has been demonstrated to partially compensate for shot-to-shot variations in ablated mass and plasma excitation characteristics [49,50,51].

Figure 8 and Figure 9 demonstrate that all calibration curves exhibited satisfactory linearity, regardless of whether carbon, aluminum, or phosphorus was used as the internal standard. The univariate calibration model with signal normalization yielded high coefficients of determination (*R^2^*) for all four analyzed elements. The concentrations of Ca, Fe, Zn and Mg in control standards *SQ2* and *SA2*, derived from intensity ratios of analyte to internal standard lines, are indicated by red triangles. These results are in good agreement with the reference values presented in Table 2 and Table 3, within the defined error margins. These results demonstrate that *LIBS* can reliably quantify specific elements under controlled laboratory conditions in pelletized pseudocereal samples, highlighting the technique’s potential while acknowledging the limited number of elements analyzed.

### 3.4. Plasma Diagnostic

To evaluate the characteristics of the laser-induced plasma and the excitation conditions relevant for elemental analysis, two key plasma parameters—the temperature (*T*) and the electron density (*N_e_*)—were determined using standard optical emission spectroscopy techniques. These parameters are fundamental indicators that directly influence the analytical performance of *LIBS*. Plasma diagnostics were conducted under optimized experimental conditions, with the laser pulse energy set to 150 mJ and the beam precisely focused on the target surface.

#### 3.4.1. Determination of Plasma Temperature Using the Boltzmann Plot Method

In contemporary studies, the Boltzmann plot method is widely applied for plasma temperature determination. In this work, Ca II ion emission lines at 315.89 nm, 317.93 nm, 373.69 nm, 393.37 nm, and 396.85 nm were employed to calculate both the plasma temperature, following established approaches reported in the literature [52,53].

Table 5 summarizes the spectroscopic parameters of the selected Ca II emission lines retrieved from the *NIST* atomic spectra database, which were employed for the determination of the plasma temperature. The applied relationship is given by the equation:(4)$$\ln\left(\frac{I_{ki}\lambda_{ki}}{g_{k}A_{ki}}\right)=-\frac{E_{k}}{K_{\beta }T}+\ln\frac{N_{e}\left(T\right)}{U\left(T\right)}$$
where *I_ki_* represents the intensity of the respective emission line, *λ_ki_* is its wavelength, *A_ki_* the transition probability, *g_k_* the statistical weight of the upper level, *E_k_* the excitation energy of that level, *K_β_* the Boltzmann constant, *U(T)*—partition function and *T* the plasma temperature. Equation (4) can be rearranged into a linear form *y = mx + b*, allowing the plasma temperature (*T*) to be derived from the slope of the fitted line, as expressed in equation:(5)$$T=-\frac{1}{K_{\beta }m}$$

Figure 10 illustrates the Boltzmann plot constructed from five Ca II ion lines belonging to the same ionization state. The plasma temperatures obtained for quinoa and amaranth samples are slightly different (10,300 K and 12,400 K, respectively; see Figure 10). Considering that a 10% difference is within the acceptable uncertainty for Boltzmann plot analysis and has negligible impact on electron density calculations, an average plasma temperature of approximately 11,000 K is used for further discussion. This value is fully consistent with the expected temperature range of *LIBS* plasma generated from organic materials (8000–15,000 K according to literature data [54] and indicates that the plasma possessed sufficient energy for effective atomization and ionization of the elements present in the samples. The observed difference between quinoa and amaranth temperatures may be related to intrinsic variations in sample composition: amaranth, with higher mineral content, could form a denser and longer-lived plasma, potentially resulting in a higher effective temperature, while quinoa, richer in organic matter, might exhibit faster plasma expansion and cooling. These observations likely reflect differences in laser–material interaction rather than experimental inconsistency, although further time-resolved or plume imaging studies would be required to confirm this behavior.

#### 3.4.2. Determination of Electron Number Density

Following the determination of the plasma temperature via the Boltzmann plot method, the electron number density (*N_e_*) was evaluated. Both plasma temperature and electron density calculations are valid under the assumption of local thermodynamic equilibrium (*LTE*). Given the 60 ms integration time of the *CCD*, these results represent time-averaged estimates, and *LTE* is treated as an approximate working assumption rather than a condition strictly satisfied throughout the plasma lifetime. Direct measurement of *N_e_* from Stark broadening of spectral lines was attempted; however, due to limited instrumental resolution and overlapping spectral features, the width of the calcium (Ca II at 396.85 nm) and hydrogen (H_α_ at 656.28 nm) lines could not be reliably used. As an initial estimate, the McWhirter criterion was applied to provide a lower bound for the electron density based on the previously determined average plasma temperature of 11,000 K [55,56]. The criterion is expressed as:(6)$$N_{e}\geq 1.6\cdot 10^{12}\sqrt{T}{\left(∆E\right)}^{3}$$
where ∆*E* is the largest energy separation between the levels of the Ca II transition at 396.85 nm. Based on the average plasma temperature, the critical electron density was estimated to be approximately 5 × 10^15^ cm^−3^, providing a lower bound required to satisfy the McWhirter criterion for *LTE* in both quinoa and amaranth samples, within the measurement uncertainty.

To obtain a more quantitative estimation of the electron number density, we used two well-resolved magnesium lines—Mg I at 285.22 nm and Mg II line at 279.55 nm [56]. These lines were chosen due to their high intensity and minimal spectral overlap, allowing reliable intensity measurements independent of Stark broadening effects. The relative population of the Mg I and Mg II excited states was determined from the measured line intensities, corrected for the respective transition probabilities (Einstein *A* coefficients) and statistical weights, according to the Boltzmann distribution [57]:(7)$$N_{j}\propto \frac{I_{\lambda }}{g_{j}A_{ij}}e^{E_{ion}/K_{\beta }T}$$
where *I_λ_* is the measured line intensity, *g_j_* the degeneracy of the upper level, *A_ij_* the Einstein coefficient, *E_j_* the energy of the upper state, *K_β_* the Boltzmann constant, and *T* the plasma temperature previously obtained from the Boltzmann plot method. The electron number density was calculated directly from the measured line intensities using the Saha equation expressed in terms of the spectral line intensities [41]:(8)$$N_{e}=\frac{I_{Mg\;I}}{I_{Mg\;II}}\frac{g_{Mg\;II}A_{Mg\;II}U_{Mg\;II}}{g_{Mg\;I}A_{Mg\;I}U_{Mg\;I}}{\left(\frac{2\pi m_{e}K_{\beta }T}{h^{2}}\right)}^{3/2}e^{\left(-\frac{E_{ion}}{K_{\beta }T}\right)}$$
where *I_Mg I_* and *I_Mg II_* are the measured line intensities, *U_Mg I_* and *U_Mg II_* are the partition functions for Mg I and Mg II, *m_e_* is the electron mass, *h* is Planck’s constant, *E_ion_* = 7.646 eV is the ionization energy of Mg I.

Using the measured line intensities of Mg I (285.22 nm) and Mg II (279.55 nm), the electron number density was estimated to be approximately 5 × 10^16^ cm^−3^. This value is roughly one order of magnitude higher than the McWhirter lower limit, indicating that the plasma conditions are consistent with the assumptions of *LTE* and supporting the applicability of the Boltzmann and Saha approaches for plasma diagnostics under the present experimental conditions. The small differences observed between quinoa and amaranth are within experimental uncertainty and, for this diagnostic parameter (electron density), do not allow us to resolve or quantify potential intrinsic variations in plasma properties.

#### 3.4.3. Numerical Analysis of the Ablated Quinoa and Amaranth Plasmas

The equilibrium composition of plasmas generated by laser ablation of quinoa and amaranth samples was computed using the numerical procedure described in Section 2.4. Calculations were performed over a temperature range of 5000–20,000 K, corresponding to the typical plasma temperatures encountered in *LIBS* experiments. The total pressure was assumed to be constant and equal to 1 bar. This pressure value is used as a standard reference condition in equilibrium plasma modeling and is not intended to represent the transient pressure evolution during *LIBS* plasma expansion; the calculations therefore provide qualitative trends in composition rather than a dynamic physical reconstruction. The initial molar compositions of quinoa and amaranth used for the plasma equilibrium calculations are summarized in Table 6. For the organic elements (H, C, O, and N), molar fractions were estimated based on the proximate composition reported in the literature [58,59], since exact elemental concentrations are not available; the percentages of proteins, lipids, carbohydrates, and fibers were converted into elemental composition. The mineral contents were obtained from *ICP*–*OES* measurements performed in this work (see Table 1). Combining these data, the molar fractions of all elements in the total composition of quinoa and amaranth were calculated and normalized to 100%.

Figure 11 and Figure 12 show the simulated equilibrium compositions of plasmas corresponding to quinoa and amaranth, respectively. Each curve represents the calculated number density of a specific element at a given ionization stage. According to the simulation, at temperatures above 14,000 K, electrons are predicted to become the most abundant species in the plasma. The modeled increase in electron density is accompanied by a corresponding rise in the calculated concentrations of singly charged ions, indicated by the dashed lines. Each figure is divided into three subplots to facilitate analysis, as displaying all elements on a single graph would result in an overly crowded and unreadable plot. The first subplot shows the main elements of the sample with significant mole fractions (above 0.1 mol %), along with electrons. The second and third subplots present elements with smaller mole fractions, arranged in descending order. According to the simulation results, at lower ionization temperatures dominant model-predicted source of electrons is potassium (blue dashed line). Between 7000 and 15,000 K, the model indicates that carbon ionization becomes the primary contributor (red dashed line), while above 15,000 K the calculated increase in electron density is further influenced by the ionization of hydrogen, oxygen, and nitrogen (black, green, and gray dashed lines). Elements with lower mole fractions have a negligible simulated effect on the electron concentration, which is why the electron density curve is shown only in the first subplot of each figure.

The electron densities estimated from Figure 11 and Figure 12, using the average plasma temperature of 11,000 K obtained in Section 3.4.1 from the calcium ion lines, are 5.2 × 10^16^ cm^−3^ for quinoa and 5.7 × 10^16^ cm^−3^ for amaranth. These values are in good agreement with the results reported in Section 3.4.2, where the electron densities were found to be approximately 5 × 10^16^ cm^−3^ for both samples. The minor differences between these estimates are within the expected uncertainty of the measurements and reflect the influence of different diagnostic approaches. It should also be noted that thermal equilibrium in the plasma exists only for a short period (~100 ns), while the *CCD* camera exposure time was much longer (60 ms). Consequently, the recorded spectral line profiles represent an integration over both equilibrium and non-equilibrium phases of the plasma, and slight variations in the calculated electron densities are therefore expected.

## 4. Conclusions

This work presents the first *LIBS* analysis of quinoa and amaranth pseudocereals. Solid samples and synthetic calibration standards were prepared, and quantitative analysis of Ca, Fe, Zn, and Mg was performed using calibration curves based on analyte-to-internal-standard intensity ratios. Distinct emission lines of Mg, Ca, Fe, K, P, Zn, Al, Sr, and Cu were observed in the recorded spectra, and limits of detection were determined. The obtained *LOD* values were sufficiently low for reliable quantification within the concentration ranges present in the samples and are consistent with typical *LIBS* performance in plant matrices. Calibration curves showed good linearity, and experimentally obtained concentrations agreed with reference *ICP*–*OES* values, supporting the reliability of the applied quantitative procedure.

Plasma diagnostics under optimized experimental conditions indicated an average plasma temperature of ~11,000 K and electron densities of ~5 × 10^16^ cm^−3^ for both quinoa and amaranth. These values are typical for *LIBS* plasma generated from organic materials and demonstrate that the applied conditions ensured effective atomization and ionization. Minor variations in the estimated parameters were within expected uncertainties and may reflect differences in sample composition. Numerical plasma simulations further supported these observations by providing model-based insight into plasma composition and ionization behavior; however, these simulations were not directly validated by time-resolved diagnostics.

Overall, the developed *LIBS* methodology demonstrates that accurate multi-element quantification of pelletized pseudocereal samples is feasible under controlled laboratory conditions. While the applicability of the approach is currently limited by the restricted set of quantified elements, the use of homogenized pressed pellets, and the absence of time-resolved plasma characterization, the results suggest that *LIBS* may have potential for application to similar cereal-like or plant-based materials, although additional studies are required to confirm this. Adaptation to rapid or field-based measurements would require alternative sample preparation strategies. Future work involving a broader analyte set, native (non-pelletized) sample forms, and temporally resolved plasma measurements would further expand the scope and robustness of this methodology. Also, the present study is based on a single commercial batch of quinoa and amaranth, and therefore the observed differences should be interpreted with caution. Future work including multiple biological replicates from different sources would strengthen the robustness and generalizability of the findings.

## Figures and Tables

**Figure 1 foods-14-04199-f001:**
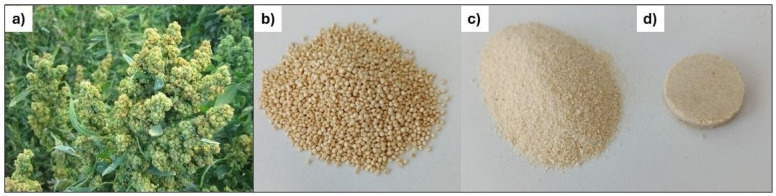
Quinoa pseudocereal used in the experiments: (**a**) photograph of a plant in its natural environment, (**b**) commercially obtained seeds, (**c**) flour produced by grinding the seeds, and (**d**) pelleted sample prepared for *LIBS* analysis.

**Figure 2 foods-14-04199-f002:**
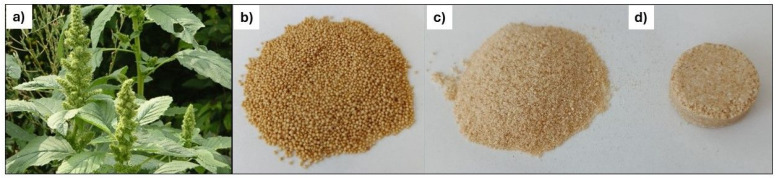
Amaranth pseudocereal used in the experiments: (**a**) photograph of a plant in its natural environment, (**b**) commercially obtained seeds, (**c**) flour produced by grinding the seeds, and (**d**) pelleted sample prepared for *LIBS* analysis.

**Figure 3 foods-14-04199-f003:**
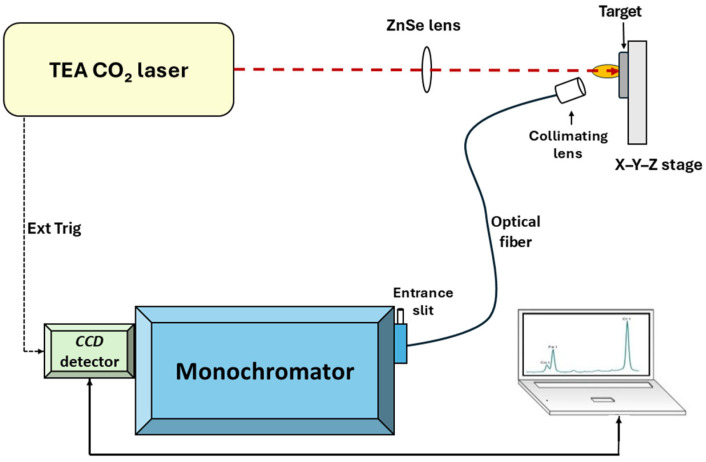
*LIBS* experimental setup.

**Figure 4 foods-14-04199-f004:**
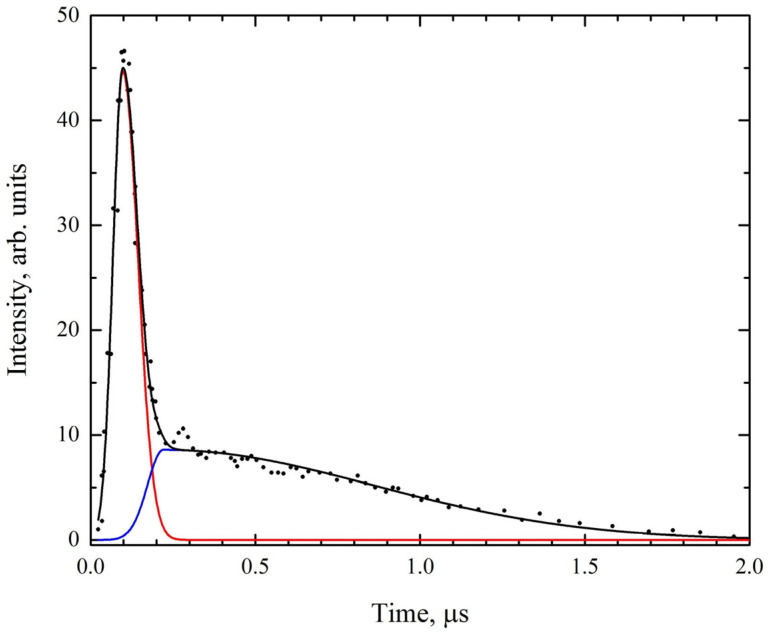
Pulse shape characteristics of the *TEA CO_2_* laser.

**Figure 5 foods-14-04199-f005:**
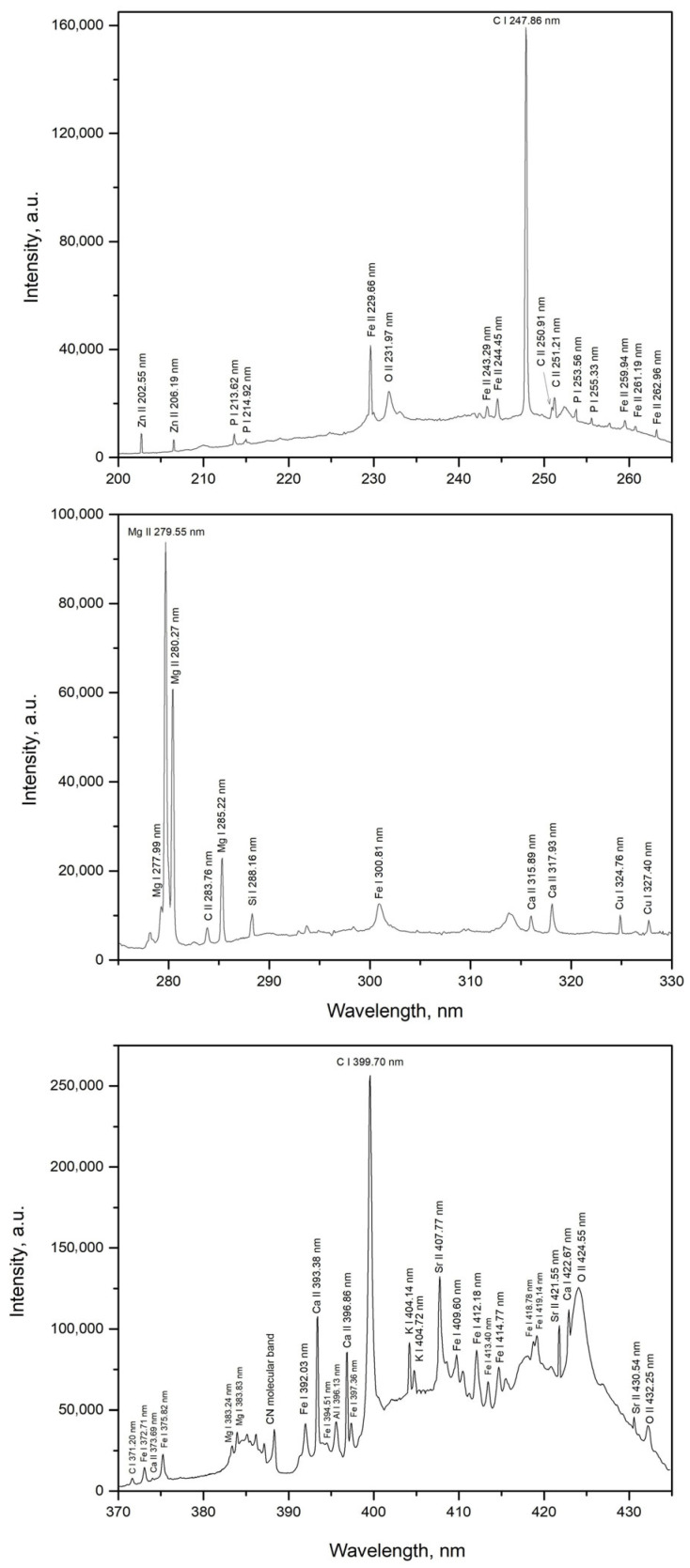
*LIBS* spectra segments of unmodified quinoa (*SQ1*).

**Figure 6 foods-14-04199-f006:**
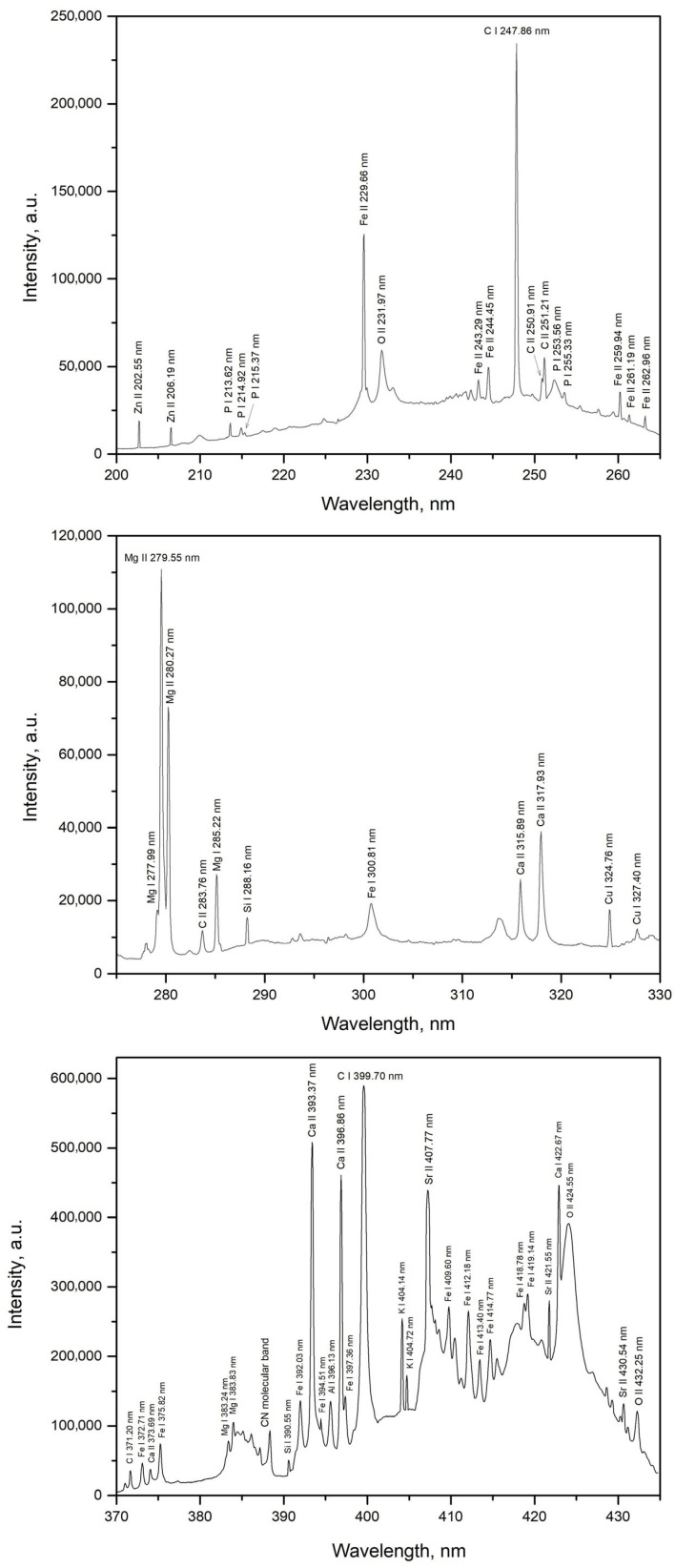
*LIBS* spectra segments of unmodified amaranth (*SA1*).

**Figure 7 foods-14-04199-f007:**
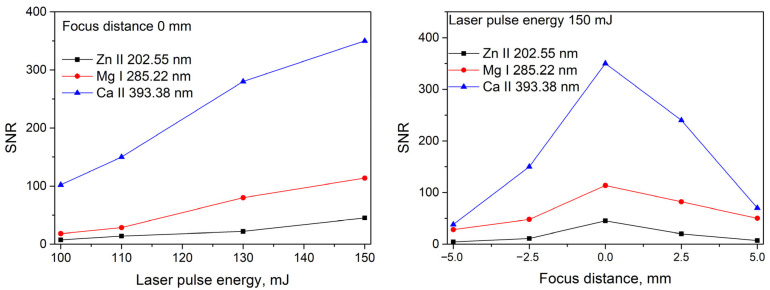
Signal-to-noise ratio (*SNR*) for the Zn II 202.55 nm, Mg I 285.22 nm, and Ca II 393.38 nm lines in the quinoa sample, as a function of laser pulse energy and focal position. Negative values in focal position indicate that the focal point of the laser beam is located behind the target surface.

**Figure 8 foods-14-04199-f008:**
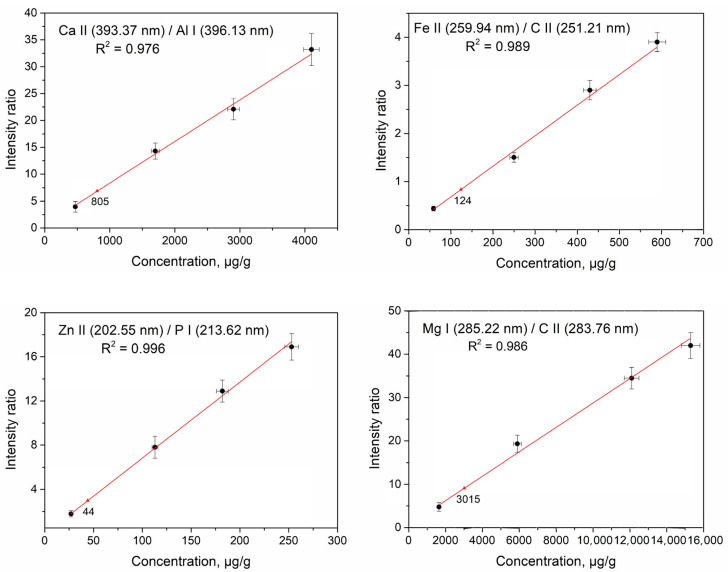
Calibration curves obtained by *LIBS* for Ca, Fe, Zn, and Mg in quinoa using intensity normalization.

**Figure 9 foods-14-04199-f009:**
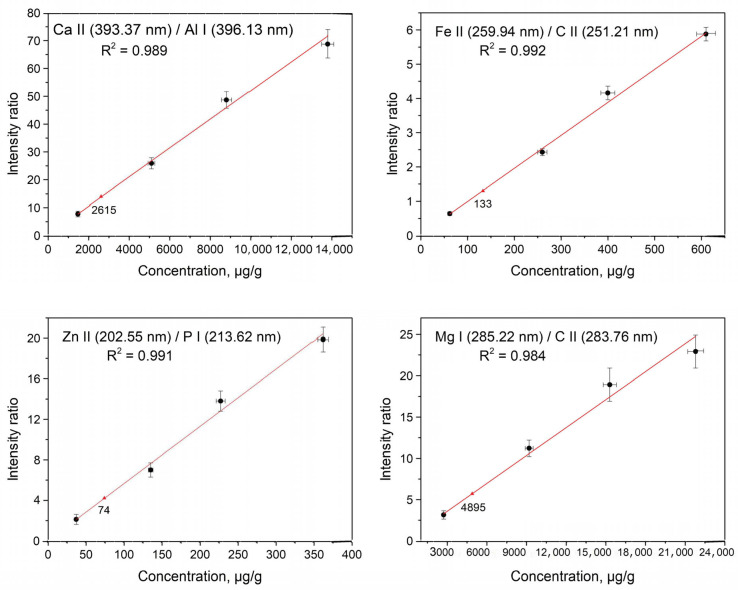
Calibration curves obtained by *LIBS* for Ca, Fe, Zn, and Mg in amaranth using intensity normalization.

**Figure 10 foods-14-04199-f010:**
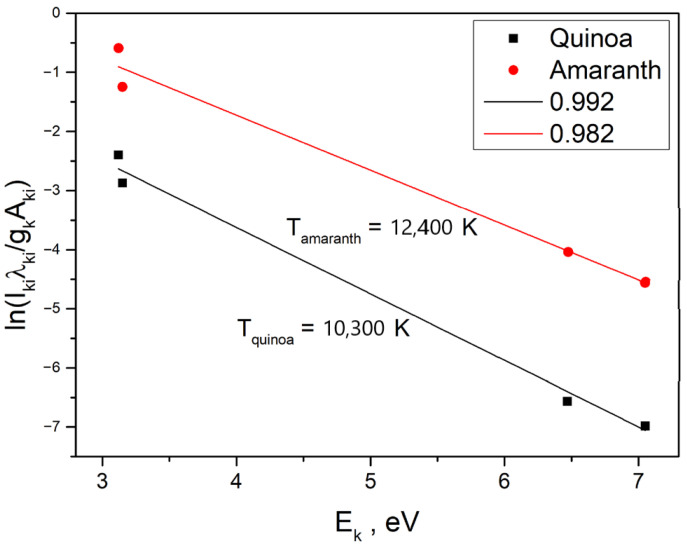
Boltzmann plots used for the determination of plasma temperature for quinoa and amaranth samples.

**Figure 11 foods-14-04199-f011:**
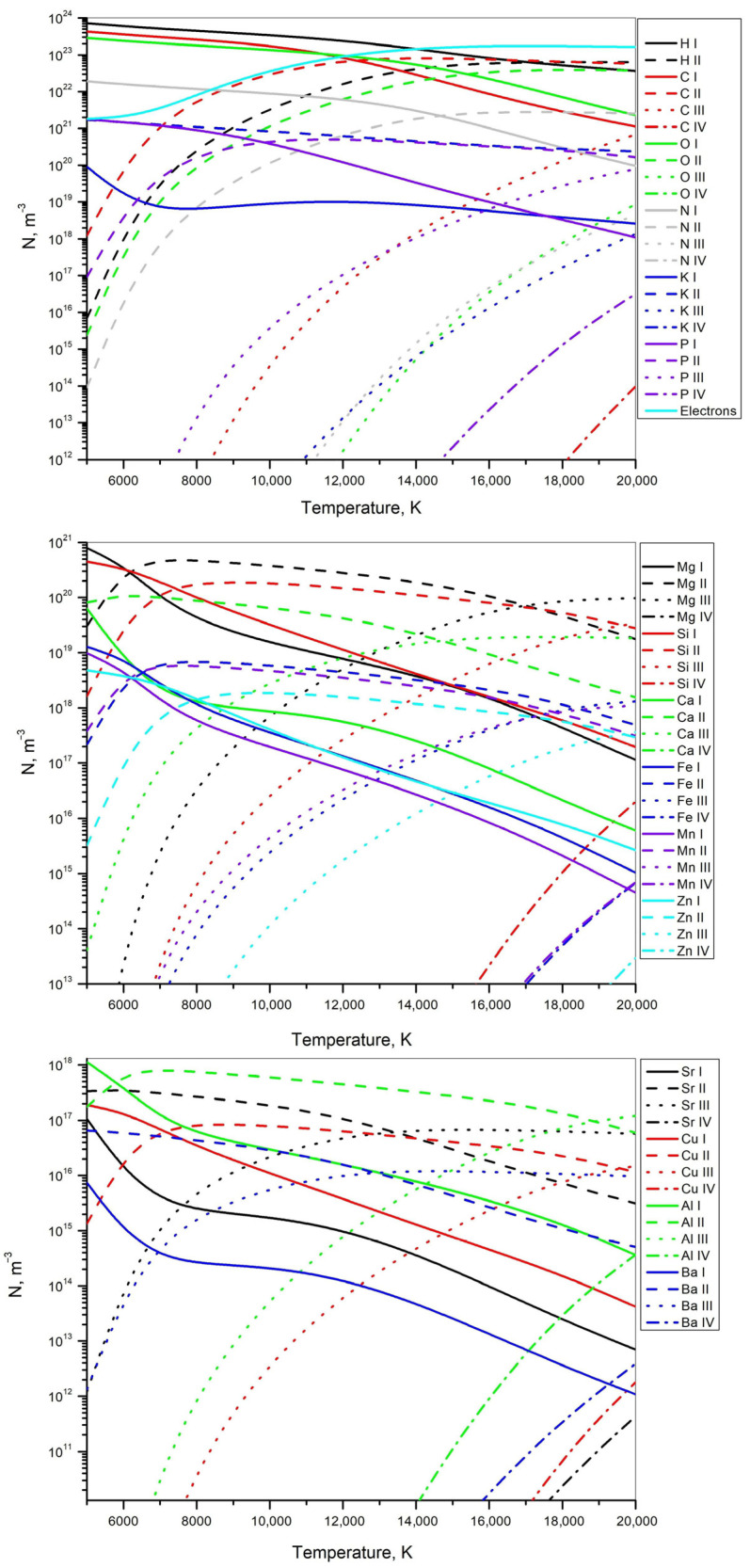
Simulated number densities of plasma constituents for quinoa at 1 bar. Line styles indicate ionization stages (I–IV), and colors represent different elements.

**Figure 12 foods-14-04199-f012:**
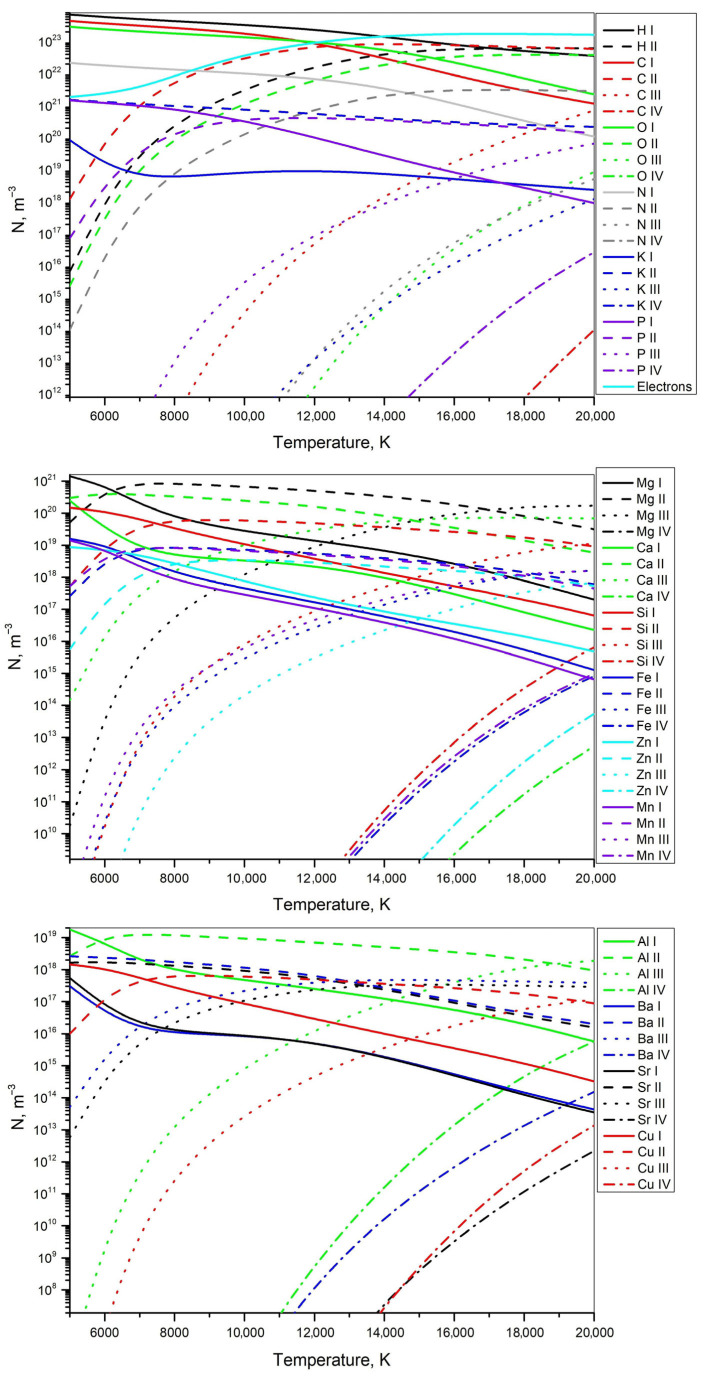
Simulated number densities of plasma constituents for amaranth at 1 bar. Line styles indicate ionization stages (I–IV), and colors represent different elements.

**Table 1 foods-14-04199-t001:** Elemental composition of quinoa and amaranth samples determined by *ICP*–*OES* (expressed in µg g^−1^).

	P	K	Mg	Ca	Fe	Zn	Si	Al	Cu	Sr
Quinoa	4435	5645	1634	468	59.2	27.2	86.1	6.44	5.06	3.12
Amaranth	5961	4875	2726	1473	61.9	37.3	28.3	16.8	4.97	2.61

**Table 2 foods-14-04199-t002:** Concentrations of selected elements (µg g^−1^) in synthetic quinoa standards prepared for *LIBS* analysis, as determined by *ICP*–*OES*.

Calibration	Concentration (µg g^−1^)			
Standard	Ca	Fe	Zn	Mg
*SQ1*	470 ± 15	59 ± 2	27 ± 1	1640 ± 50
*SQ2*	820 ± 30	122 ± 4	46 ± 2	2960 ± 70
*SQ3*	1700 ± 60	250 ± 10	113 ± 3	5900 ± 210
*SQ4*	2900 ± 90	430 ± 15	182 ± 6	12,100 ± 400
*SQ5*	4100 ± 120	590 ± 20	253 ± 7	15,300 ± 500

**Table 3 foods-14-04199-t003:** Concentrations of selected elements (µg g^−1^) in synthetic amaranth standards prepared for *LIBS* analysis, as determined by *ICP*–*OES*.

Calibration	Concentration (µg g^−1^)			
Standard	Ca	Fe	Zn	Mg
*SA1*	1470 ± 50	62 ± 2	37 ± 1	2730 ± 50
*SA2*	2660 ± 80	131 ± 4	73 ± 2	4830 ± 160
*SA3*	5100 ± 170	260 ± 10	135 ± 3	9200 ± 300
*SA4*	8800 ± 250	400 ± 15	227 ± 6	15,300 ± 500
*SA5*	13,800 ± 300	610 ± 20	362 ± 7	21,800 ± 600

**Table 4 foods-14-04199-t004:** Limits of detection (*LOD*) obtained by *LIBS* analysis of quinoa and amaranth samples.

Element	Wavelength (nm)	*SNR quinoa*	*c_quinoa_* (µg g^−1^)	*LOD_quinoa_* (µg g^−1^)	*SNR Amaranth*	*c_amaranth_* (µg g^−1^)	*LOD_amaranth_* (µg g^−1^)
P I	213.62	35	4435	380	51	5961	350
K I	404.14	339	5645	50	325	4875	45
Mg I	285.22	114	1634	43	137	2726	60
Ca II	393.38	350	468	4	330	1473	14
Fe II	259.94	107	59.2	2	99	61.9	2
Zn II	202.55	45	27.2	2	51	37.3	2
Si I	288.16	65	86.1	4	57	28.3	2
Al I	396.13	38	6.44	0.5	63	16.8	1
Cu I	324.76	68	5.06	0.3	61	4.97	0.3
Sr II	407.77	54	3.12	0.2	47	2.61	0.2

**Table 5 foods-14-04199-t005:** Spectroscopic parameters of Ca II lines used for plasma temperature determination.

Ion	Wavelength (nm)	*A_ki_* (10^8^ s^−1^)	*g_k_*	*E_k_* (eV)
Ca II	315.89	3.1	4	7.05
Ca II	317.93	3.6	6	7.05
Ca II	373.69	1.7	2	6.47
Ca II	393.37	1.5	4	3.15
Ca II	396.85	1.4	2	3.12

**Table 6 foods-14-04199-t006:** Initial molar composition of quinoa and amaranth used for plasma equilibrium calculations.

Element	Quinoa (mol %)	Amaranth (mol %)	Source/Basis
H	49.14	49.7	Calculated from literature data
C	29.4	29.5	Calculated from literature data
O	19.8	19.4	Calculated from literature data
N	1.3	1.5	Calculated from literature data
K	0.121	0.0487	ICP–OES (this work)
P	0.120	0.1	ICP–OES (this work)
Mg	5.65 × 10^−2^	0.1	ICP–OES (this work)
Si	3.07 × 10^−3^	2.83 × 10^−3^	ICP–OES (this work)
Ca	9.84 × 10^−3^	0.0147	ICP–OES (this work)
Al	8.9 × 10^−3^	1.68 × 10^−3^	ICP–OES (this work)
Fe	8.92 × 10^−3^	6.19 × 10^−3^	ICP–OES (this work)
Zn	3.27 × 10^−3^	3.73 × 10^−3^	ICP–OES (this work)
Cu	1.3 × 10^−3^	4.97 × 10^−4^	ICP–OES (this work)
Mn	6.99 × 10^−4^	1.0 × 10^−3^	ICP–OES (this work)
Sr	3.0 × 10^−4^	1.5 × 10^−4^	ICP–OES (this work)
Ba	5.0 × 10^−6^	2.0 × 10^−5^	ICP–OES (this work)

## Data Availability

Data supporting the findings of this study are available from the corresponding author upon reasonable request.

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
