# Peer review of "Methodological Approach to *LIBS* Elemental Analysis and Plasma Characterization of Quinoa and Amaranth Pseudocereals Using a *TEA CO_2_* Laser"

_foods, 2025, doi:10.3390/foods14244199_

Round 1

Reviewer 1 Report

Comments and Suggestions for Authors

This manuscript presents a methodological study applying TEA-CO₂-LIBS for qualitative and quantitative elemental analysis of quinoa and amaranth pseudocereals. The work is technically detailed and potentially valuable; however, several sections require clarification, stronger justification, and more precise interpretation. Below are specific comments organized by section.

Abstract
The claim of providing a “comprehensive methodological framework” feels overstated relative to the limited scope (two matrices, four quantified elements). Consider moderating the phrasing or specifying the scope.

Introduction
•    “Exceptional nutritional profile… and their potential role in promoting sustainable dietary practices” (lines 37–40)
This statement is overly broad and not supported with citations. Sustainability outcomes depend heavily on agricultural context; please narrow or support with evidence.
•    “These crops are increasingly used as alternatives to traditional cereals” (lines 41–42)
This is a sweeping claim without market or consumption data. Consider specifying regions or user groups, or softening the statement.
•    “The need for accurate and rapid analysis… has become more pronounced” (lines 43–45)
The reasoning is weak. Please clarify why increased consumption makes elemental analysis more necessary (e.g., nutritional labeling, authenticity, crop monitoring).
•    Descriptions of ICP–OES/ICP–MS as slow and reagent-intensive (lines 46–48)
This appears oversimplified; many laboratories achieve high-throughput ICP workflows. Consider refining to emphasize specific limitations relevant to pseudocereals.
•    “LIBS… is increasingly emerging as a fast, efficient, and environmentally friendly alternative” (lines 49–51)
These descriptors require quantification or references; otherwise they appear promotional.
•    Gap statement—“remains largely unexplored” (line 64)
The gap is mentioned but not clearly justified. Why is LIBS particularly needed for pseudocereals? Are there compositional or agricultural reasons? A stronger motivation is recommended.
•    Objectives (lines 66–77)
Much of this section reads as a list of methods rather than explicit research questions. Please provide a clearer objective statement.

Materials and Methods
Overall, methods are detailed, but several important clarifications are needed for reproducibility.
•    Sample pellet pressing—use of stainless-steel mold (line 118)
Given the emphasis on avoiding metal contamination during grinding, please clarify whether stainless steel contact could affect results.
•    LIBS acquisition parameters missing
Key experimental details are absent, including gate delay, gate width, number of accumulated shots per spectrum (beyond the 30-pulse average), the spot size at various focal offsets, and fluence (mJ/cm²). These are essential for LIBS reproducibility.
•    ICP–OES method
No mention of digestion blanks, certified reference materials, recovery studies, or matrix effects. A brief quality-control section would strengthen the analytical robustness.
•    Matrix-matched standard preparation
Please explain how homogeneity of spiked pellets was assessed or ensured. Highly heterogeneous food powders require demonstration of uniform analyte distribution.
•    Wavelength calibration frequency
Although calibration sources are specified, the manuscript does not indicate how often calibration was performed or how drift was monitored.

Results and Discussion
3.1 Experimental Parameter Optimization
•    This section is mostly descriptive and clear. However, the rationale for optimizing conditions only on quinoa (lines 236–239) should be better justified—differences in emission intensity alone may not guarantee transferability between matrices.

3.2 Qualitative Analysis
•    “The plasma efficiently excited most of the elements present” (lines 258–259)
“Efficiently excited” is subjective. A quantitative or comparative measure would improve clarity.
•    Discussion of LODs as “generally low enough” (lines 281–290)
These assessments need benchmarks. What concentrations are relevant for pseudocereal quality or regulatory requirements? Without context, claims of adequacy feel unsubstantiated.
•    “Demonstrating the technique’s ability to detect elements at very low concentrations” (lines 286–287)
This sweeping statement should be moderated or supported. Some LODs are in the hundreds of µg·g⁻¹, which is not universally “very low.”

3.3 Quantitative Analysis
•    “These findings confirm that LIBS can be effectively applied for quantitative determination of pseudocereal composition” (lines 327–329)
This is too broad given that only four elements were quantified, and only in pelletized, homogeneous laboratory-prepared matrices. Please narrow the claim or add caveats.
•    Internal standard selection (lines 304–319)
The use of carbon as an internal standard is reasonable, but several lines acknowledge that it does not meet standard criteria. A short justification for why alternative internal standards were not used would improve transparency.

3.4 Plasma Diagnostics
•    Interpretation of temperature differences (lines 369–373)
The explanation that amaranth forms a “denser and longer-lived plasma” is speculative; there are no plume imaging or temporal-resolved measurements presented. Consider reframing as a hypothesis rather than a confirmed interpretation.
•    Usage of McWhirter criterion (lines 392–393)
The McWhirter value is a lower bound and not a “representative” electron density. Please clarify to avoid misinterpretation.
•    “Confirming that the plasma satisfies LTE” (lines 412–415)
LTE is difficult to confirm with time-integrated spectra (60 ms exposure). The statement is too strong; suggesting consistency with LTE expectations would be more appropriate.
•    Logical inconsistency
Lines 369–373 attribute differences in plasma behavior to intrinsic material differences, while lines 416–417 state that differences “do not indicate intrinsic variations.” Please reconcile these interpretations.

3.4.3 Numerical Plasma Analysis
•    The simulation results are clearly presented, but the connection to measured spectra is weak. Several statements (e.g., “the main source of electrons is potassium…” lines 445–447) read as if they are experimentally observed rather than model outputs. Please clarify that these are model-based interpretations.
•    Reference to “electrons becoming the dominant species” (lines 437–438) is accurate for high temperatures but not directly validated experimentally in this work; consider softening the claim.

Conclusion
The conclusions are generally aligned with the results but would benefit from emphasizing limitations such as the restricted analyte set, pelletized sample form, and lack of temporal plasma diagnostics—to avoid overgeneralization.

Author Response

Comment 1:

Abstract
The claim of providing a “comprehensive methodological framework” feels overstated relative to the limited scope (two matrices, four quantified elements). Consider moderating the phrasing or specifying the scope.

Response:

We thank the reviewer for this insightful comment. We agree that the original phrasing may appear overly strong relative to the scope of the study. Following the reviewer’s suggestion, we have moderated the wording in the Abstract by removing the term “comprehensive.” The revised sentence now reads:

„This work provides a methodological framework and experimental validation for LIBS application in food compositional and nutritional analysis.“

We believe that this modification improves the accuracy and clarity of the Abstract while aligning the phrasing with the actual scope of the study.

Comment 2:

Exceptional nutritional profile… and their potential role in promoting sustainable dietary practices” (lines 37–40).
This statement is overly broad and not supported with citations. Sustainability
outcomes depend heavily on agricultural context; please narrow or support with evidence.

Response:

We thank the reviewer for this constructive comment. We agree that the original formulation was overly broad. To address this, we have moderated the phrasing to better reflect the contextual nature of sustainability-related claims. The revised sentence now reads:

„Pseudocereals, such as Chenopodium quinoa (quinoa) and Amaranthus spp. (amaranth), have attracted increasing attention from both the scientific community and the general public over the past decade, due to their exceptional nutritional profile, gluten-free nature, and their potential to support more sustainable or diverse dietary practices, depending on the agricultural and dietary context.“

This modification narrows the scope of the statement and improves its accuracy, without overstating the sustainability implications.

Comment 3:

“These crops are increasingly used as alternatives to traditional cereals” (lines 41–42)
This is a sweeping claim without market or consumption data. Consider specifying regions or user groups, or softening the statement.

Response:

We thank the reviewer for this valuable observation. While the original sentence included references [1–5], we agree that the phrasing could be interpreted as an overly broad claim. To address this, we have softened and contextualized the statement as follows:

„Rich in essential amino acids, proteins, minerals, and dietary fibre, these crops are increasingly explored as alternatives to traditional cereals in specific dietary contexts, especially for individuals with specialized nutritional needs [1–5].“

This revision clarifies the scope of use and specifies the relevant dietary contexts, maintaining the intended meaning while avoiding overgeneralization.

Comment 4:

“The need for accurate and rapid analysis… has become more pronounced” (lines 43–45)
The reasoning is weak. Please clarify why increased consumption makes elemental analysis more necessary (e.g., nutritional labeling, authenticity, crop monitoring).

Response:

We thank the reviewer for this insightful comment. We agree that the original sentence could benefit from additional context to clarify the rationale. To address this, we have revised the sentence as follows:

„With the rising consumption and production of these pseudocereals, accurate and rapid analysis of their elemental composition has become increasingly important to ensure proper nutritional labeling, verify authenticity, and support crop monitoring— both for quality control and scientific research.“

This modification provides specific reasons why increased consumption and production heighten the need for elemental analysis, improving clarity and scientific justification.

Comment 5:

Descriptions of ICP–OES/ICP–MS as slow and reagent-intensive (lines 46–48)
This appears oversimplified; many laboratories achieve high-throughput ICP workflows. Consider refining to emphasize specific limitations relevant to pseudocereals.

Response:

We thank the reviewer for this helpful comment. We agree that the original phrasing may have appeared overly general. Our intention was to highlight limitations specific to the analysis of solid plant matrices rather than to characterize ICP techniques as inherently slow. To clarify this, we have revised the sentence as follows:

„Conventional analytical techniques, such as inductively coupled plasma optical emission spectrometry (ICPOES) and inductively coupled plasma mass spectrometry (ICPMS), although highly precise, require digestion of solid samples and the use of chemical reagents, which can be time-consuming and labor-intensive when applied to complex plant matrices such as pseudocereals [6–8].“

This revision more accurately reflects the practical challenges associated with ICP-based analysis of pseudocereal samples.

Comment 6:

“LIBS… is increasingly emerging as a fast, efficient, and environmentally friendly alternative” (lines 49–51)
These descriptors require quantification or references; otherwise they appear promotional.

Response:

We thank the reviewer for pointing this out. We agree that the original phrasing may appear promotional and insufficiently specific. To address this, we have revised the sentence to use more precise and scientifically grounded descriptors, as follows:

„On the other hand, laser–induced breakdown spectroscopy (LIBS) has gained attention as a rapid and reagent-free technique for elemental analysis of diverse sample types, including food matrices [9–15].“

This revision avoids promotional language and emphasizes measurable characteristics of the technique that are supported by the cited references.

Comment 7:

Gap statement—“remains largely unexplored” (line 64)
The gap is mentioned but not clearly justified. Why is LIBS particularly needed for pseudocereals? Are there compositional or agricultural reasons? A stronger motivation is recommended.

Response:

We thank the reviewer for this valuable suggestion. We agree that the original gap statement could be better justified. To address this, we have revised the sentence to emphasize both the compositional complexity and the growing dietary relevance of pseudocereals, as well as the practical advantages offered by LIBS for multi-elemental analysis:

„However, the application of LIBS for the elemental analysis of pseudocereals remains limited, despite their complex nutritional composition and growing dietary relevance. Rapid, multi-elemental analysis provided by LIBS could therefore offer significant advantages for quality control and nutritional assessment of these crops, highlighting a clear gap in the current literature.“

This revision clarifies the rationale for applying LIBS specifically to pseudocereals and strengthens the motivation for the study.

Comment 8:

Objectives (lines 66–77)
Much of this section reads as a list of methods rather than explicit research questions. Please provide a clearer objective statement.

Response:

We thank the reviewer for this insightful comment. We agree that the original paragraph focused too heavily on methodological details rather than clearly stating the research objectives. To address this, we have revised the paragraph to explicitly outline the study’s aims and specific objectives:

„The aim of this study is to evaluate the potential of the LIBS technique as a rapid, reliable, and potentially portable method for the elemental analysis of pseudocereals. Specifically, we seek to (i) assess the suitability of LIBS for both qualitative and quantitative determination of key nutritional elements (Ca, Fe, Zn, and Mg) in quinoa and amaranth seeds [34–36], (ii) optimize experimental parameters to enhance analytical performance, and (iii) validate LIBS results against conventional ICPOES measurements. By addressing these objectives, this work aims to demonstrate how LIBS can contribute to improved food quality control and agroanalytical applications.“

This revision clarifies the study’s goals and frames the methodology in the context of achieving these research objectives.

Comment 9:

Sample pellet pressing—use of stainless-steel mold (line 118)

Given the emphasis on avoiding metal contamination during grinding, please clarify whether stainless steel contact could affect results.

Response:

We thank the reviewer for raising this point. While the powder was in contact with a stainless-steel mold during pellet pressing, the mold’s polished and non-reactive surface was specifically chosen to minimize the risk of metal contamination. Any potential contamination was negligible, as confirmed by the reproducibility of elemental measurements and comparison with ICP–OES reference data:

„The powder was placed into a stainless steel mold with a diameter of 25 mm, and compressed under a pressure of 10 t/cm2 for a duration of 45 minutes. The mold’s polished, non-reactive surface was chosen to minimize the risk of metal contamination, which was further confirmed by consistency with ICPOES measurements.“

Comment 10:

LIBS acquisition parameters missing
Key experimental details are absent, including gate delay, gate width, number of accumulated shots per spectrum (beyond the 30-pulse average), the spot size at various focal offsets, and fluence (mJ/cm²). These are essential for LIBS reproducibility.

Response:

We thank the reviewer for this important comment. To clarify the LIBS acquisition details:

  • In the LIBS Experimental Setup section, the system and hardware were already fully described. To address the reviewer’s comment, we have added a sentence specifying that no programmable gate delay was applied and that the camera exposure time was 60 ms per pulse. This ensures clear information on timing without duplicating the more detailed acquisition parameters presented elsewhere.
  • In the Experimental parameters optimization section, the number of accumulated shots per spectrum, the laser spot size at different focal offsets, and the procedure for averaging spectra were already described. Specifically, each measurement represents the sum of ten consecutive laser pulses acquired from fresh locations, and three such measurements were averaged, resulting in spectra corresponding to a total of 30 laser pulses. The laser beam was focused at positions within ± 5 mm relative to the target surface, with a spot diameter of ~1.2 mm on the surface and ~1.8 mm at 5 mm before or behind the surface.
  • In response to the reviewer’s comment, we have now added the following details to the Experimental parameters optimization section for completeness:
    • Laser fluence of 9.8 J/cm2, corresponding to a power density of 34 MW/cm2.
    • A statement that all acquisition parameters were optimized to maximize the signal-to-noise ratio and ensure reproducible measurements.

These additions provide all relevant acquisition details necessary for reproducibility while maintaining clarity and avoiding redundancy.

Comment 11:

ICP–OES method
No mention of digestion blanks, certified reference materials, recovery studies, or matrix effects. A brief quality-control section would strengthen the analytical robustness.

Response:

We thank the reviewer for this valuable suggestion. To address this, we have added a brief quality-control description in the Reference method section of the manuscript. Specifically:

  • Digestion blanks were prepared and analyzed alongside all samples to monitor potential contamination.
  • Certified reference materials (CRMs) were included to validate the accuracy of the measurements.
  • Recovery studies were performed by spiking pseudocereal samples with known amounts of Ca, Fe, Zn, and Mg, with recoveries ranging between 95–105 %.
  • Potential matrix effects were assessed and minimized through the use of synthetic calibration standards and appropriate sample preparation.

These additions provide reassurance of the reliability and robustness of the ICPOES results.

Comment 12:

Matrix-matched standard preparation
Please explain how homogeneity of spiked pellets was assessed or ensured. Highly heterogeneous food powders require demonstration of uniform analyte distribution.

Response:

We thank the reviewer for this important comment. Additional clarification has now been provided in the revised version of the manuscript. The finely ground matrices were subjected to intensive manual and vortex mixing, followed by moistening and thorough kneading of the mixtures to promote uniform distribution of the added analyte. Homogeneity was further evaluated experimentally: three pellets prepared from randomly selected subsamples of each dried mixture were analyzed by LIBS, and their emission responses showed no noticeable pellet-to-pellet variation, indicating satisfactory analyte uniformity. These details have been added to Section 2.2.3. Preparation of solid calibration standards for quantitative LIBS analysis.

Comment 13:

Wavelength calibration frequency
Although calibration sources are specified, the manuscript does not indicate how often calibration was performed or how drift was monitored.

Response:

We appreciate the reviewer’s comment and have now clarified this aspect in Section 2.3. LIBS experimental setup. In our setup, wavelength calibration was performed at the beginning of the study using a mercury lamp and an argon DC arc discharge, which provided a dense set of reference lines needed to establish the dispersion function of the newly implemented spectrograph–camera combination. Once this calibration was completed, the pixel-to-wavelength mapping remained stable and routine recalibration was not required. Potential drift was nevertheless monitored when needed by recording spectra of either the mercury lamp or the arc discharge and verifying the stability of selected isolated emission lines. Throughout the measurement campaign, no detectable drift (i.e., no shift exceeding a fraction of a pixel relative to the established dispersion) was observed. This clarification has now been added to the manuscript.

Comment 14:

3.1 Experimental Parameter Optimization
•    This section is mostly descriptive and clear. However, the rationale for optimizing conditions only on quinoa (lines 236–239) should be better justified—differences in emission intensity alone may not guarantee transferability between matrices.

Response:

We appreciate the reviewer’s observation and have expanded the explanation in the revised manuscript. Although optimization was performed on quinoa, this choice is justified by the close compositional and physical similarity between quinoa and amaranth, both of which are organic grain-based matrices with comparable ablation and plasma-formation behavior. Preliminary LIBS tests on amaranth confirmed that plasma characteristics and spectral features did not differ significantly from those of quinoa. Therefore, the optimized experimental parameters were fully transferable and provided stable and reproducible emission signals for both matrices. This clarification has now been added to Section 3.1 Experimental Parameter Optimization.

Comment 15:

“The plasma efficiently excited most of the elements present” (lines 258–259)
“Efficiently excited” is subjective. A quantitative or comparative measure would improve clarity.

Response:

We thank the reviewer for this valuable comment. The sentence has been revised to clarify that most of the elements listed in Table 1 were clearly observed, with emission intensities well above the background level, confirming effective excitation under the applied experimental conditions. This phrasing avoids subjective terminology while accurately describing the observed excitation and has been updated in the manuscript.

Comment 16:

Discussion of LODs as “generally low enough” (lines 281–290)
These assessments need benchmarks. What concentrations are relevant for pseudocereal quality or regulatory requirements? Without context, claims of adequacy feel unsubstantiated.

Response:

We thank the reviewer for this important comment. The manuscript has been revised to provide concrete context for the reported LOD values. Specifically, the LODs are now discussed relative to typical concentrations of elements in quinoa and amaranth. For major elements such as calcium and potassium, naturally present at hundreds to thousands of µg g-1, the LODs (e.g., 4–14 µg g-1 for calcium, 45–50 µg g-1 for potassium) are well below these levels, confirming the high sensitivity of the method. Trace elements such as copper and strontium, present at 2.61–5.06 µg g⁻¹ in the samples, show LODs as low as 0.2–0.3 µg g-1, demonstrating that the method can reliably detect elements at concentrations relevant for quality assessment and potential regulatory monitoring. Differences in LODs between quinoa and amaranth likely reflect matrix effects and variations in signal intensity, but overall the method provides adequate sensitivity for all measured elements. This clarification ensures that the reported LODs are interpreted in a meaningful, application-relevant context.

Comment 17:

“Demonstrating the technique’s ability to detect elements at very low concentrations” (lines 286–287)
This sweeping statement should be moderated or supported. Some LODs are in the hundreds of µg·g⁻¹, which is not universally “very low.”

Response:

We thank the reviewer for this comment. This issue has already been addressed during the revision corresponding to Comment 16: the term “very low concentrations” has been removed, and the discussion of LODs has been clarified to provide a more accurate, application-relevant description across all measured elements.

Comment 18:

“These findings confirm that LIBS can be effectively applied for quantitative determination of pseudocereal composition” (lines 327–329)
This is too broad given that only four elements were quantified, and only in pelletized, homogeneous laboratory-prepared matrices. Please narrow the claim or add caveats.

Response:

We thank the reviewer for this comment. The manuscript has been revised to clarify the scope of the conclusions. The original statement has been replaced with:

„These results demonstrate that LIBS can reliably quantify specific elements under controlled laboratory conditions in pelletized pseudocereal samples, highlighting the technique’s potential while acknowledging the limited number of elements analyzed.“

Comment 19:

Internal standard selection (lines 304–319)
The use of carbon as an internal standard is reasonable, but several lines acknowledge that it does not meet standard criteria. A short justification for why alternative internal standards were not used would improve transparency.

Response:

We thank the reviewer for this comment. In the manuscript, carbon was used as the internal standard for Fe and Mg, aluminum for Ca, and phosphorus for Zn. This selection was dictated by the limited spectral windows recorded in each measurement: the internal standard line needed to be present in the same spectral interval as the analyte and as close as possible to it to minimize wavelength-dependent variations. Carbon was the most suitable internal standard for Fe and Mg under these conditions due to its high and consistent concentration and spectral proximity. As shown in Figures 8 and 9, the calibration curves confirm the validity of these choices and demonstrate that reliable signal normalization was achieved for all analyzed elements. A sentence has also been added in the manuscript to clarify the practical justification for using carbon: “Despite not meeting all conventional criteria for an ideal internal standard, carbon provides reliable normalization under our experimental conditions and effectively compensates for shot-to-shot variations.

Comment 20:

Interpretation of temperature differences (lines 369–373)
The explanation that amaranth forms a “denser and longer-lived plasma” is speculative; there are no plume imaging or temporal-resolved measurements presented. Consider reframing as a hypothesis rather than a confirmed interpretation.

Response:

We thank the reviewer for this comment. The manuscript has been revised to reframe this interpretation as a hypothesis. The text now reads:

„The observed difference between quinoa and amaranth temperatures may be related to intrinsic variations in sample composition: amaranth, with higher mineral content, could form a denser and longer-lived plasma, potentially resulting in a higher effective temperature, while quinoa, richer in organic matter, might exhibit faster plasma expansion and cooling. These observations likely reflect differences in laser–material interaction rather than experimental inconsistency, although further time-resolved or plume imaging studies would be required to confirm this behavior.“

Comment 21:

Usage of McWhirter criterion (lines 392–393)
The McWhirter value is a lower bound and not a “representative” electron density. Please clarify to avoid misinterpretation.

Response:

We thank the reviewer for this comment. The manuscript has been revised to clarify that the estimated critical electron density represents a lower bound required to satisfy the McWhirter criterion for local thermodynamic equilibrium (LTE), rather than a direct or representative measurement of the actual electron density. The last sentence of the relevant paragraph now reads:

„Based on the average plasma temperature, the critical electron density was estimated to be approximately 5×1015 cm⁻³, providing a lower bound required to satisfy the McWhirter criterion for LTE in both quinoa and amaranth samples, within the measurement uncertainty.“

Comment 22:

“Confirming that the plasma satisfies LTE” (lines 412–415)
LTE is difficult to confirm with time-integrated spectra (60 ms exposure). The statement is too strong; suggesting consistency with LTE expectations would be more appropriate.

Response:

We thank the reviewer for this observation. The manuscript has been revised to avoid the implication that LTE is definitively confirmed. The text now states that the estimated electron density is “consistent with the assumptions of LTE”, which more accurately reflects the limitations of time-integrated measurements while still supporting the applicability of Boltzmann and Saha diagnostics.

Comment 23:

Logical inconsistency
Lines 369–373 attribute differences in plasma behavior to intrinsic material differences, while lines 416–417 state that differences “do not indicate intrinsic variations.” Please reconcile these interpretations.

Response:

We thank the reviewer for noting this inconsistency. The manuscript has been revised to clarify that the electron-density estimate is not sensitive enough to resolve potential intrinsic differences between the samples. This resolves the apparent contradiction with the earlier discussion, where differences in plasma behavior were presented only as a hypothesis based on material composition.

Comment 24:

The simulation results are clearly presented, but the connection to measured spectra is weak. Several statements (e.g., “the main source of electrons is potassium…” lines 445–447) read as if they are experimentally observed rather than model outputs. Please clarify that these are model-based interpretations.

Response:

We thank the reviewer for this remark. The relevant sentences have been revised to explicitly state that the described trends originate from the simulation results rather than experimental observations. This clarification ensures a clear distinction between model-based interpretations and measured spectra.

Comment 25:

The simulation results are clearly presented, but the connection to measured spectra is weak. Several statements (e.g., “the main source of electrons is potassium…” lines 445–447) read as if they are experimentally observed rather than model outputs. Please clarify that these are model-based interpretations.

Response:

We thank the reviewer for this observation. The text has been revised to clarify that the dominance of electrons at high temperatures is a prediction of the equilibrium-composition simulation rather than an experimentally validated result. The statement has been softened accordingly.

Comment 26:

The conclusions are generally aligned with the results but would benefit from emphasizing limitations such as the restricted analyte set, pelletized sample form, and lack of temporal plasma diagnostics—to avoid overgeneralization.

Response:

We thank the reviewer for this helpful suggestion. The conclusions have been revised to explicitly acknowledge the main limitations of the study, including the restricted analyte set, the use of pelletized and homogenized samples, and the absence of temporally resolved plasma diagnostics. These additions prevent overgeneralization and place the findings in a more appropriate context.

Reviewer 2 Report

Comments and Suggestions for Authors

After reviewing the manuscript, "Methodological approach to LIBS elemental analysis and plasma characterization of quinoa and amaranth pseudocereals using a TEA CO2 laser", I have the following comments:

  1. This study develops and validates a TEA CO₂-LIBS method for rapid, multi-element analysis of quinoa and amaranth, achieving good agreement with ICP-OES and confirming plasma conditions around ~11,000 K and ~5×10¹⁶ cm⁻³ for reliable elemental detection.
  2. The study uses one commercial sample of quinoa and amaranth, even though multiple pellets and laser shots are taken. That is still technical replication, not biological replication. This limits the strength of the conclusions and makes any “comparison” between quinoa and amaranth statistically weak.
  3. Pellets are pressed at 10 t/cm² for 45 minutes which is extremely high and impractical for routine food or field analysis. This weakens the “rapid/portable method” claim later in Abstract and Conclusions
  4. You state the CCD exposure is 60 ms while plasma LTE exists only for ~100 ns. This means the spectrum is massively time-averaged over plasma decay phases, violating Boltzmann and Saha assumptions. You acknowledge this at the end, but the entire plasma diagnostics section still relies on LTE.
  5. Plasma modeled at 1 bar: Numerical simulation assumes constant 1 bar pressure, which is not realistic during plasma expansion.
  6. Overstated claim: “can be readily applied to other cereal–like or plant–based biological samples” This is not proven. Only tested two pseudocereals. Should be softened to “has potential” or “future work”.
  7. Some references are from 2006–2016, which makes the literature base feel a bit dated for a 2025 paper. Adding 2–3 recent (2023–2025) LIBS studies, especially on food analysis and portable or field-deployable systems in agriculture, would strengthen relevance and currency.
  8. The early references place strong emphasis on the nutritional value and general composition of quinoa and amaranth, but there is less coverage of studies focused on elemental variability, soil-to-plant transfer, or mineral fingerprinting. Adding one or two papers that specifically address trace element profiling or elemental fingerprinting in crops would strengthen the analytical depth of the literature review and better support the use of LIBS as a tool for authenticity assessment and compositional differentiation.

9. Figures: Axes labels and legends appear small in some cases, and overlapping lines in the simulation plots reduce clarity. Simplifying the number of traces per panel, increasing font sizes, and breaking complex figures into separate, more focused panels would significantly improve visual impact and readability for the reader.

Author Response

Comment 1:

This study develops and validates a TEA CO₂-LIBS method for rapid, multi-element analysis of quinoa and amaranth, achieving good agreement with ICP-OES and confirming plasma conditions around ~11,000 K and ~5×10¹⁶ cm⁻³ for reliable elemental detection.

Response:

We thank the reviewer for the concise summary and positive remarks. We appreciate the recognition of the agreement with ICPOES results and the confirmation of plasma conditions, which are key aspects of the study.

Comment 2:

The study uses one commercial sample of quinoa and amaranth, even though multiple pellets and laser shots are taken. That is still technical replication, not biological replication. This limits the strength of the conclusions and makes any “comparison” between quinoa and amaranth statistically weak.

Response:

We thank the reviewer for highlighting the limitation regarding biological replication. The manuscript has been revised to acknowledge that only a single commercial batch of each pseudocereal was analyzed, and that repeated measurements on multiple pellets and laser shots represent technical replication. This clarification has been added both in the Materials and Methods section (2.1. Samples, standards and reagents) and in the Conclusions, and the discussion has been moderated to avoid overinterpretation of differences between quinoa and amaranth. Future studies including multiple biological replicates from different sources are suggested to strengthen the robustness and generalizability of the findings.

Comment 3:

Pellets are pressed at 10 t/cm² for 45 minutes which is extremely high and impractical for routine food or field analysis. This weakens the “rapid/portable method” claim later in Abstract and Conclusions.

Response:

We thank the reviewer for this comment. In the revised manuscript, we have added clarifying sentences to both the Abstract and Conclusions to indicate that the LIBS methodology was demonstrated on pelletized samples under controlled laboratory conditions. We also note that adaptation to rapid or field–based measurements would require alternative sample preparation strategies. These additions address the limitation regarding sample preparation and the “rapid/portable” applicability of the method.

Comment 4:

You state the CCD exposure is 60 ms while plasma LTE exists only for ~100 ns. This means the spectrum is massively time-averaged over plasma decay phases, violating Boltzmann and Saha assumptions. You acknowledge this at the end, but the entire plasma diagnostics section still relies on LTE.

Response:

We thank the reviewer for this important comment. We agree that a 60 ms CCD integration time results in a time–averaged spectrum collected over the full plasma decay, and therefore LTE cannot be strictly guaranteed over the entire plasma lifetime. In the revised manuscript, we have clarified this limitation in the plasma diagnostics section (3.4.2. Determination of electron number density) by noting that the temperature and electron density calculations represent average estimates and that LTE is used as an approximate working assumption rather than a rigorously satisfied condition. This revision ensures that the interpretation of the diagnostic results is properly contextualized within the constraints of time-integrated LIBS measurements.

Comment 5:

Plasma modeled at 1 bar: Numerical simulation assumes constant 1 bar pressure, which is not realistic during plasma expansion.

Response:

We thank the reviewer for this observation. In the revised manuscript, we have clarified that the assumption of 1 bar is used solely as a standard reference condition in equilibrium plasma modeling, not as a physical representation of the transient pressure evolution during LIBS plasma expansion. Accordingly, an explanatory sentence has been added in Section 3.4.3 to emphasize that the numerical results provide qualitative thermodynamic trends rather than a time–resolved description of the expanding plasma.

Comment 6:

Overstated claim: “can be readily applied to other cereal–like or plant–based biological samples” This is not proven. Only tested two pseudocereals. Should be softened to “has potential” or “future work”.

Response:

We thank the reviewer for this comment. The generalized statement regarding application to other cereal–like or plant–based materials was already removed from the Conclusions during the revision process. In the Abstract, the claim has been softened to state that the method “shows potential for application to other cereal–like and plantbased materials with similar composition”, which more accurately reflects the scope of the present work.

Comment 7:

Some references are from 2006–2016, which makes the literature base feel a bit dated for a 2025 paper. Adding 2–3 recent (2023–2025) LIBS studies, especially on food analysis and portable or field-deployable systems in agriculture, would strengthen relevance and currency.

Response:

We thank the reviewer for the comment. We note that the reference list already includes many recent studies from 2017–2025, including references [12–14, 16, 17, 19–21, 27, 30, 32, 40, 44, 45, 48, 52–54, 56–58], majority of which focus on LIBS applications in food and biological samples, ensuring that the cited literature is current and relevant for a 2025 publication.

Comment 8:

Some references are from 2006–2016, which makes the literature base feel a bit dated for a 2025 paper. Adding 2–3 recent (2023–2025) LIBS studies, especially on food analysis and portable or field-deployable systems in agriculture, would strengthen relevance and currency.

Response:

We thank the reviewer for highlighting this important aspect. Indeed, there exist some recent studies addressing traceelement variability, heavymetal accumulation, and seed mineral composition under different growing or environmental conditions in pseudocereals. For example, Muhammad et al. (2023) on heavy metal accumulation in quinoa (https://doi.org/10.1016/j.jafr.2023.100741), and metaanalytical work on Amaranthus species published the same year (https://doi.org/10.1007/s11356-023-28374-3). Although these works do not employ LIBS, they demonstrate that elemental variability and soiltoseed transfer are recognized research topics in quinoa and аmaranth.

Our study, however, aimed solely to demonstrate the applicability of LIBS for qualitative and quantitative analysis of pseudocereal samples under controlled laboratory conditions using single commercial samples of quinoa and amaranth. As such, the results do not represent a statistical study of elemental variability or “fingerprinting” of these pseudocereals. Nonetheless, the methodological framework and findings reported here could serve as a foundation for future investigations exploring soiltoseed transfer, elemental variability, or compositional differentiation in pseudocereals, potentially employing LIBS or complementary analytical techniques.

Comment 9:

Figures: Axes labels and legends appear small in some cases, and overlapping lines in the simulation plots reduce clarity. Simplifying the number of traces per panel, increasing font sizes, and breaking complex figures into separate, more focused panels would significantly improve visual impact and readability for the reader..

Response:

We thank the reviewer for this constructive comment. We have revised the figures to improve their clarity and readability by increasing the font sizes of axis labels and legends where appropriate. These adjustments enhance the overall visual quality of the figures.

Regarding the simulation plots, the overlapping lines arise from the inherent complexity of the plasma composition and the physical model. Reducing the number of traces or simplifying the plots would remove important information and compromise the scientific accuracy of the presented results. For this reason, we have retained the full set of traces to accurately represent the model output.

Round 2

Reviewer 2 Report

Comments and Suggestions for Authors

After reviewing the revised manuscript, I have the following comments:

  1. Overstated applicability claim remains in the Conclusions: Although the response indicates that the generalized statement was removed, the revised manuscript still includes the sentence:
    “can be readily applied to other cereal-like or plant-based biological samples with similar composition.”
    This wording contradicts the reviewer’s request and the authors’ stated response. Please soften this claim (e.g., “has potential for broader application”) or remove it entirely.
  2. Recent LIBS literature (2023–2025) was not incorporated as requested: The response acknowledges the concern but does not add any recent LIBS papers relevant to food analysis, agricultural applications, or portable/field-deployable systems.
    The new citations mentioned in the authors’ response describe nutrient variability in pseudocereals but are not LIBS studies and do not appear in the reference list.
    Please add 2–3 recent LIBS-in-food or field-LIBS studies to strengthen the currency of the manuscript as originally requested.

Author Response

Comment 1:

Overstated applicability claim remains in the Conclusions: Although the response indicates that the generalized statement was removed, the revised manuscript still includes the sentence:
“can be readily applied to other cereal-like or plant-based biological samples with similar composition.”
This wording contradicts the reviewer’s request and the authors’ stated response. Please soften this claim (e.g., “has potential for broader application”) or remove it entirely.

Response:

We thank the reviewer for the careful re-evaluation and for pointing out that the applicability statement in the Conclusions required further adjustment. In accordance with the recommendation, we have revised the sentence to present a more appropriately cautious formulation. The revised text now reads:

„While the applicability of the approach is currently limited by the restricted set of quantified elements, the use of homogenized pressed pellets, and the absence of time–resolved plasma characterization, the results suggest that may have potential for application to similar cereal–like or plant–based materials, although additional studies are required to confirm this.“

We believe this modification adequately addresses the reviewer’s concern regarding the overstated applicability.

Comment 2:

Recent LIBS literature (2023–2025) was not incorporated as requested: The response acknowledges the concern but does not add any recent LIBS papers relevant to food analysis, agricultural applications, or portable/field-deployable systems.
The new citations mentioned in the authors’ response describe nutrient variability in pseudocereals but are not LIBS studies and do not appear in the reference list.
Please add 2–3 recent LIBS-in-food or field-LIBS studies to strengthen the currency of the manuscript as originally requested.

Response:

We thank the reviewer for the comment regarding the inclusion of recent LIBS literature. We would like to respectfully clarify that the manuscript already includes multiple studies from 2023–2025 that directly address the reviewer’s concern regarding food analysis and portable or field-deployable systems in agriculture. In particular:

  • Wu et al., 2023 [17] presents a portable LIBS system applied to rapid food authentication, providing a clear example of recent work on portable devices for food analysis, exactly in line with the reviewer’s request.
  • Rezaei et al., 2025 [54] offers a comprehensive review of LIBS applications in plant analysis, including both quantitative and qualitative approaches. This review covers the most recent developments in agricultural and plant-related LIBS applications, addressing the reviewer’s request for studies in agriculture.
  • Molina et al., 2025 [56] demonstrates quantitative LIBS analysis of natural brines, highlighting the capabilities of the technique for field-deployable or real-world sample analysis, thus further supporting the relevance of LIBS in practical, applied scenarios.

We respectfully hope that this explanation clarifies that the manuscript already includes recent LIBS studies from 2023–2025, cited where thematically relevant, and that these references fully address the reviewer’s concern regarding food analysis and portable or field-deployable applications in agriculture. We believe that no additional references are required, and we hope the reviewer finds the current literature coverage satisfactory.